# Cost-Effectiveness Analysis of Pneumococcal Vaccines in the Pediatric Population: A Systematic Review

**DOI:** 10.3390/healthcare12191950

**Published:** 2024-09-29

**Authors:** Nam Xuan Vo, Huong Lai Pham, Uyen My Bui, Han Tue Ho, Tien Thuy Bui

**Affiliations:** 1Faculty of Pharmacy, Ton Duc Thang University, Ho Chi Minh City 700000, Vietnam; h1900276@student.tdtu.edu.vn (H.L.P.); h1900341@student.tdtu.edu.vn (U.M.B.); h2100032@student.tdtu.edu.vn (H.T.H.); 2Faculty of Pharmacy, Le Van Thinh Hospital, Ho Chi Minh City 700000, Vietnam; bttien.ths.tcqld23@ump.edu.vn

**Keywords:** cost-effectiveness analysis, pneumococcal vaccine, pediatric, systematic review

## Abstract

**Objectives:** Pneumococcal disease, caused by *Streptococcus pneumoniae*, is the leading cause of mortality in children worldwide. The tremendous direct cost of hospital admissions and significant indirect costs from productivity loss contribute considerably to its economic burden, with vaccination being the only efficient protection against the illness. Our study aims to summarize the cost-effectiveness of the pneumococcal conjugate vaccine (PCV) implemented in the pediatric population. **Methods:** Employing the online databases PubMed, Embase, and Medline, we looked for economic evaluations from 2018 until March 2024. The Incremental Cost-Effectiveness Ratios (ICER) and Quality-Adjusted Life Years (QALY) were the primary outcomes for measuring the cost-effectiveness of PCVs. A 28-item CHEERS 2022 checklist was applied to assess the quality of the collected studies. **Results:** Of the 16 papers found, 9/16 discussed the lower-valent vaccines (PCV13, PCV10) and 7/16 examined the higher-valent vaccines (PCV20, PCV15). PCV13 and PCV10 involved greater costs and generated more QALY compared to no vaccination. Both PCV15 and PCV20 averted substantial healthcare costs and yielded greater quality of life than PCV13. Additionally, PCV20 was a dominant strategy compared to PCV15. **Conclusions:** Utilizing PCV13 is a very cost-effective option compared to not getting vaccinated. Transitioning from PCV13 to PCV20 would result in higher QALY gain and more cost-saving than switching to PCV15.

## 1. Introduction

*Streptococcus pneumoniae* is the causative pathogen of pneumococcal disease (PD), which is currently the most widespread infectious illness worldwide [1,2,3,4,5,6,7,8]. There are two forms of PD: pneumococcal invasive disease (IPD) and non-invasive pneumococcal disease [9]. Despite a relatively low occurrence, IPD is the most severe presentation, involving meningitis, bacteremia, and sepsis [10]. The mortality rate is substantial, reaching 10% for meningitis and 15% for bacteremia, and survivors may have serious aftereffects [3]. Conversely, non-invasive pneumococcal disease includes more prevalent but milder conditions such as otitis media, sinusitis, and pneumonia [11,12,13].

Pneumococcus infections can strike persons of any age. However, those with underlying medical conditions, young children under 2, and adults over 65 are probably the most susceptible [10]. The annual mortality rate from pneumococcal disease among children is approximately one million, with the majority of these deaths occurring in developing nations [8,14]. Pneumonia significantly contributes to this burden, evidenced by its association with approximately 11–20 million severe hospitalized cases among children under 5 years of age [15]. Recent estimates from the World Health Organization (WHO) indicated that in 2019, pneumonia was responsible for the demise of 740,180 children under 5 years old, constituting 14% of all pediatric deaths within this age [16]. Notably, the most vulnerable regions were South Asia (2500 cases per 100,000 children) and West and Central Africa (1620 cases per 100,000 children) [17]. In Ethiopia, acute respiratory infections are the leading cause of mortality among children under five, comprising 18% of deaths within this age group. Pneumonia contributes explicitly to 16.4% of all deaths in the country, making Ethiopia the highest pneumonia-related mortality rate among Sub-Saharan African nations [17].

Given that *S. pneumoniae* stands out as the most common cause of community-acquired-pneumonia (CAP) in children <5 years of age [18], the treatment expenses were tremendous. The implementation of pneumococcal conjugate vaccines (PCVs) via infant immunization has demonstrated success in mitigating the prevalence of pneumococcal infections [19,20,21,22]. In the USA, a review of the literature in 2021 regarding the cumulative 20-year effect of PCVs among children under the age of five showed that PCVs helped to reduce 91% of IPD incidences between 1997 and 2019 [23]. The first vaccine circulated was PCV7 in the USA in the year 2000, serving the pediatric population [23,24]. The prevalence of *S. pneumoniae* infections has decreased significantly since the dissemination and implementation of PCV13, which provided protection against a broader range of serotypes than PCV7 [25]. Additionally, annual pneumonia hospitalizations decreased by 66–79%, from 110,000 cases among children under five years old [23]. In England and Wales, the implementation of PCV13 has led to a reduction of over 50% in the overall incidence of IPDs [25].

The economic burden of disease is enormous: healthcare costs were USD 13.7 billion each year, while societal costs were USD 14.3 billion worldwide [26]. In Europe, the aggregate medical bills associated with CAP were approximated at EUR 10.1 billion annually, with about one-third of these expenditures allocated to indirect costs [27]. The primary contributor to the considerable financial burden of the illness was the hospitalization cost [26,27,28]. A 19-year evaluation of Canadian healthcare expenses following the introduction of PCV13 in infants found that admission to the hospital accounted for 92% of overall costs, which amounted to CAD 7.25 billion (2018 value) [28]. In a low-income country like Nigeria, the surveillance discovered that one-third of the households suffered from hospitalization charges that exceeded 25% of their monthly income, emphasizing the burden of significant treatment costs [26]. Not to mention considerable indirect costs from productivity losses, including premature disability and fatalities in children (USD 3.1 billion in 2004) and days of work loss for taking care of their child (USD 914 million in 2004) [19].

Nevertheless, the burden of PD persists due to the emergence of non-vaccine serotypes (NVT), a phenomenon known as serotype replacement [14,15,27,29,30]. The ongoing rise in the occurrence of NVT serotypes, explicitly 3 and 19A, may counteract the positive impact of the decreased prevalence of vaccine serotypes (VTs) subsequent to the implementation of PCV13 [25,31]. Currently, the development of higher-valent vaccines is under way to address this issue of NVT, including PCV15, which has recently gained approval for use in infants and adults in Europe, the US, and Canada, and PCV20, which has been approved for adult use in Europe and the US [15,32]. Given that vaccine costs remain relatively high and accessibility remains a concern, it is imperative to assess the cost-effectiveness of these vaccines. Such data are essential for governments to allocate resources effectively based on informed decisions. Consequently, we aim to conduct a systematic review focusing on the cost-effectiveness of authorized vaccines currently available on the market to elucidate the prevailing trends in this regard.

## 2. Materials and Methods

### 2.1. Search Strategy

Our research was carried out using the PRISMA 2020 checklist [33]. We conducted a comprehensive search for economic analyses pertaining to pneumococcal vaccines from 2018 to 2024 using electronic sources, including PubMed, Medline, and Embase. The last article was retrieved on March 2024. Key phrases including “pneumococcal disease”, “PCV”, “pediatric”, and “cost-effectiveness analysis” were used to search publications. The main code used in the search process proceeded as follows: (pneumococcal OR pneumococcal disease) AND (cost-effectiveness OR CEA OR Economic Evaluation OR EE OR cost-utility analysis OR CUA OR cost-benefit analysis OR CBA OR cost-minimization analysis OR CMA) AND (pneumococcal conjugate vaccine OR PCV) AND (pediatric OR Children).

### 2.2. Selection Process

The inclusion criteria for the economic evaluations are as follows: (1) the study falls within the scope of economic analysis, encompassing cost-effectiveness analysis (CEA), cost-utility analysis (CUA), or cost-benefit analysis (CBA); (2) the investigation centers on the pediatric population (under 18 years of age); (3) specific pneumococcal vaccines are compared; (4) pertinent information regarding health outcomes, such as incremental cost-effectiveness ratio (ICER), quality-adjusted life years (QALY), or disability-adjusted life years (DALY) is provided; (5) a clear determination regarding the cost-effectiveness of the intervention is presented; (6) full-text accessibility is ensured; (7) articles are published in English.

Conversely, articles will be disqualified if they meet any of the following exclusion criteria: (1) the research does not fall within the scope of economic analysis; (2) participants are adults; (3) the comparison of pneumococcal vaccines is not clearly specified; (4) data regarding health outcomes are ambiguous; (5) no definitive conclusion regarding cost-effectiveness is drawn; (6) full-text accessibility is unavailable; (7) reports are not written in English.

### 2.3. Data Extraction

The chosen reports will then be summarized by extracting the following details: the first author, publication year, country, type of economic evaluation, target clinical outcomes, comparison, study, model approach, time horizon, discount rate, currency, perspective, vaccine coverage, vaccine schedule, herd effect inclusion in base-case analysis, funding source, health outcomes, and sensitivity analysis. Our primary outcome is ICER (incremental cost-effectiveness ratio), which can be defined as the ratio of the incremental costs to the incremental effectiveness achieved between two interventions [34]:$$\mathrm{ICER}=\frac{Cost\;\left(intervention\right)-Cost\;(comparator)}{Effectiveness\;\left(intervention\right)-Effectiveness\;(comparator)}$$

The effectiveness of health benefits can be measured in terms of QALY (Quality-Adjusted Life Years) or DALY (Disability-Adjusted Life Years). All of the ICERs were retrieved within the base-case analysis of each cost-effectiveness evaluation.

The study includes two types of costs: direct and indirect. Direct costs include vaccination costs and healthcare costs (such as hospitalization expenses or treatment costs), while indirect costs encompass non-medical costs, such as caregivers’ productivity loss when caring for their sick children.

If feasible, cost-effectiveness thresholds (CE threshold) were also derived from economic evaluation to compare the ICERs. This is the utmost monetary value that a healthcare decision-maker or society is willing to expend in exchange for one unit of health benefit [35]. In general, a cost-effectiveness threshold is established to identify interventions that exhibit comparatively or exceptionally favorable value for money [36]. GDP-based thresholds or WTP-based thresholds may be applied. Following the World Health Organization’s Choosing Interventions that are Cost–Effective project (WHO-CHOICE) suggestion, if the cost per disability-adjusted life years (DALY) avoided was less than the national annual ×1 GDP per capita of the country, the intervention is deemed “very cost-effective”, whereas the range within 2–3 times GDP per capita is considered as “cost-effective”, and then if the cost exceeds ×3 GDP per capita it is called “not cost-effective” [36]. Willingness-to-pay (WTP) is the highest possible level of money the government is willing to pay in exchange for a health benefit. This threshold will determine whether or not a specific intervention is financially worthwhile to invest in. An intervention is thought to be “cost-effective” if the ICERs lie within the WTP range; conversely, if they go above the WTP, it is deemed “not cost-effective” [37].

The secondary outcome is sensitivity analysis (SA), which encompasses both deterministic sensitivity analysis (DSA) and probabilistic sensitivity analysis (PSA). Its aim is to assess uncertainty or examine the robustness of specific model outcomes under plausible ranges of key parameters. DSA employs a tornado diagram to visually represent the impact of the most significant parameter inputs, dynamically adjusting the value of a single economic model parameter at a time (e.g., case fatality rate, costs, or indirect effects). In contrast, PSA involves selecting each parameter from 1000 iterations of Monte Carlo simulation. This PSA enables the estimation of the disparity of the ICER when multiple inputs are varied simultaneously.

### 2.4. Articles Quality Assessment

To verify our selected economic analysis technique, we used the Consolidated Health Economic Evaluation Reporting Standards (CHEERS) 2022 tool checklist [38]. The checklist, which consists of 28 elements, evaluates the essential data that must be provided for a typical economic analysis. Each item can be given one of three values: “1” for a fully completed response, “0.5” for a partially completed response, and “0” for information that was not relevant or was not applicable. Then, we ranked the quality of reports based on the total score; those above 21 were deemed as high quality, reports scoring between 14–24 were classified as moderate quality, and the ones less than 14 were considered poor quality [39]. All of the articles were reviewed separately by two independent researchers.

## 3. Results

### 3.1. Studies Selection Process

Our selection process is summarized in Figure 1. A total of 434 articles were identified from different electronic databases, of which PubMed accounted for 188 papers, Cochrane covered three articles, and the rest were from Embase. Following the removal of duplicates, we decomposed 409 papers by applying inclusion and exclusion criteria to the title and abstract screening. Consequently, we were left with 18 reports, from which we extracted data by accessing the full text. Two studies were eliminated during the process: one from Wang et al. [40] compared immunization schedules, and the other from Eythorsson et al. [41] could not articulate health outcomes but instead broke down clinical outcomes into distinct categories to evaluate cost-effectiveness. In the end, sixteen papers met the requirements for data analysis.

### 3.2. Characteristics of Included Studies

The overall information of the chosen evaluation is demonstrated in Table 1. The majority of collected articles presented CEA, with 13/16; in 3/16 articles, CUA was conducted [42,43,44] while 1/16 used CBA [42]. The Markov model was the most widely used approach, accounting for 50% of the studies [42,43,45,46,47,48,49,50]. Time horizon varied from 1 year to 100 years (lifetime), with the length of 10 years being the most applied period (8/16 articles) [44,45,48,49,50,51,52,53]. The discount rate also fluctuated from 1.50% to 7%, with the level of 3% being the most popular rate (9/16 publications) [42,43,46,47,49,50,51,54,55]. Each of the articles can apply more than one perspective; payer [42,44,45,48,53,54,56,57] and societal perspective [42,45,46,47,48,50,52] were both recorded in 8/16 papers. Additionally, vaccine uptake levels could be different depending on how long one country had been applying PCVs, ranging from 70% to 100%. Most of the studies were sponsored by pharmaceutical companies such as Pfizer (68.7%, 11/16 studied) [42,44,45,46,49,50,52,53,54,57] and Merck (12.5%, 2/16 papers) [47,48]; one other was funded by international organizations such as WHO, GAVI, and Melinda Gates Foundation (6.2%, 1/16 paper) [55].

### 3.3. Quality Assessment

16/16 articles were considered good quality, with the lowest score being 21.5/28 [44]. Details of quality grading following the CHEERS 2022 statement are presented in Table 2. Most of the articles did not meet the information for item 21, “Approach to engagement with patients and others affected by the study”, and item 25, “Effect of engagement with patients and others affected by the study”.

### 3.4. Cost-Effectiveness Related Data

#### 3.4.1. Lower-Valent Pneumococcal Vaccine

The incremental effect of lower-valent pneumococcal vaccine is summarized in Table 3, including PCV13 and PCV10. The majority of studies that conducted PCV13 analysis focused on Asia [42,46,51,54,56]. Compared to no vaccination, implementing PCV13 would increase total cost (mainly driven by direct cost) and provide greater QALY [42,46,51,54,55,56]. All of the ICERs were below ×1 GDP per capita, indicating PCV13 is a cost-effective strategy [42,43,46,54,56]. In China, including herd effect would reduce the total cost and extend QALY, thus leading to much lower ICER per QALY gained (79,304 CNY vs. 3777 CNY; cost inflated to the year 2015) demonstrating a very cost-effective choice, compared to the herd-effect-excluded scenario [54]. In addition, PCV10 was reported to be cost-effective in Egypt and Bhutan, compared to no vaccination, with the ICERs lower than x1 GDP per capita [42,43]. In contrast, PCV10 was not cost-effective due to ICER exceeding the WTP of 160,000 TBH/QALY in Thailand (cost inflated to the year 2018) when excluding the herd effect [46].

Regardless of the vaccine schedule, PCV13 was estimated to be more cost-saving than utilizing PCV10 [42,44,52], with higher QALY gain and greater savings of direct costs (from hospitalization expenses).

#### 3.4.2. Higher-Valent Pneumococcal Vaccine

The results of ICER and cost reduction of the newest vaccines are illustrated in Table 4. Compared to PCV13, all of the higher valency PCVs were reported to be dominant strategies [45,47,48,49,50,57]. These CEAs about the newest licensed vaccines took place in developed countries such as the USA, UK, Japan, Canada, and Germany. Implementing PCV15 resulted in higher vaccine costs, but the averted pneumococcal treatment costs offset these [47,48,57]. Consequently, using PCV15 led to higher QALY, higher LYs, and more cost-saving in terms of a 3 + 1 schedule. In the UK, with the willingness-to-pay threshold (WTP) of 20,000 GBP/QALY, choosing PCV15 in the 1 + 1 schedule was more cost-effective than PCV13 in the 1 + 1 schedule but not in PCV15 2 + 1 schedule, as the ICER was greater than WTP (313,229 GBP/QALY vs. 20,000 GBP/QALY) [57].

Similarly, PCV20 was dominant compared to PCV13 in 4/4 analyses [45,49,50,57]. Though it required investment to introduce the vaccine, the productivity loss reduction and the prevented medical cost outweighed it significantly. Considering the 1 + 1 schedule, PCV20 provided higher QALY gain (28,096 vs. 361), LYs gain (23,165 vs. 262), and more cost reduction (GBP 459 million vs. GBP −1 million) than PCV15 when compared with PCV13 [57] (cost was inflated to the year 2021).

On the other hand, PCV20 was reported to be a dominant option over PCV15 regardless of the vaccine schedule [45,49,50,53,57], which was less costly and had higher QALY and LYs.

### 3.5. Sensitivity Analysis

The results of sensitivity analysis in vaccines with lower valency is demonstrated in Table 5. When comparing PCV13 and PCV10 to no vaccination, the most prominent parameters that affect ICERs are discount rate [42,43,56] and incidence rate [42,54,55,56]. The result from PSA indicated that 100% of iterations focused on the northeast quadrant. It appeared that utilizing PCV13 would be more expensive but would also lead to a more significant improvement in quality of life compared to not using any immunization [46,51,55].

The results of DSA and PSA in higher-valent pneumococcal vaccines are presented in Table 6. It can be seen that the indirect effect was one of the parameters that affected the cost value [45,49,50] in DSA. In addition, the ICER of these PCVs was sensitive to vaccination cost [49]. However, in PSA results, PCV20 was shown to be a dominant intervention compared to both PCV13 and PCV15, as all of the simulations focused on the southeast quadrant, which indicated PCV20 was less costly and more effective than PCV13 [45,49,50,53]. The same pattern was observed in PCV15 vs. PCV13; the majority of the simulations demonstrated positive ICERs [47,48], indicated that PCV15 would be a more economical option to prevent pneumococcal disease compared to PCV13.

## 4. Discussion

Based on our research findings, employing PCV13 or PCV10 demonstrated enhanced cost-effectiveness and augmented quality-adjusted life years (QALY) in comparison to the absence of vaccination. The majority of ICERs were below half of the Gross Domestic Product (GDP) per capita [42,43,51], or falling within the WTP threshold [46,55], suggesting a very cost-effective option. Furthermore, despite the higher monetary investment necessary to manage PCV13, it has the potential to decrease the direct cost, particularly hospitalization costs, significantly. According to a 10-year cohort study in India, introducing the PCV13 vaccination program in India would cost USD 35 million (cost was inflated to the year 2017) [51]. However, the vaccine helped prevent 25,134,220 pneumococcal cases, which was equivalent to USD 51.6 million in healthcare cost reduction and averted 920,000 DALY compared to getting no immunization [51]. Similarly, the economic evaluation conducted in Egypt over a 100-year period by implementing PCV13 resulted in a saving of USD 0.88 in direct costs and a rise of 0.0462 QALY [42]. This resulted in an ICER of USD 926 per QALY gain (costs were inflated to 2016), which was equal to half of the country’s GDP per capita of USD 3479 [42]. This makes PCV13 a more cost-effective option, regardless of whether the herd effect is taken into account or the vaccine schedule is considered [42,44,52]. Employing PCV13 on a global scale would have the greatest impact in Africa and Asia, which averted annually 8.68 million DALY (95% Cl: 3.99 million to 16.8 million) and 3.88 million DALY (95% Cl: 1.75 million to 6.78 million), respectively [55]. The global cost of launching the vaccine was USD 15.5 billion. However, implementing PCV13 in countries that are not yet implementing the PCV program would require an investment of one-third of the global cost, which was USD 4.42 million [55].

Studies in Egypt, the Philippines, and South Africa demonstrated that PCV13 contributed higher cost reductions and extended QALY compared to PCV10 [42,44,52]. Bhutan and India are classified as low-middle income countries, whereas Egypt is considered a developing country with a per capita income that exceeds the threshold for qualifying for Gavi financial support. Among the various presentations of pneumococcal disease, acute otitis media (AOM) emerges as the foremost contributor to direct costs, primarily attributable to expenses incurred from antibiotic prescriptions, owing to its elevated incidence. As elucidated in the investigation conducted by Pichichero et al. (2023) on the burden of AOM, regions such as those within South and Southeast Asia, characterized by their status as low- and middle-income countries (LMICs), harbor the highest prevalence of the predominant strains responsible for AOM, largely due to the constrained distribution of PCVs [58]. Consequently, the adoption of PCV13 emerges as the most prudent strategy for LMICs, facilitating optimal allocation of resources and fostering improvements in quality of life.

On the other hand, our findings indicated that the most recent licensed vaccines, PCV15 and PCV20, demonstrated dominance when compared to the widely utilized PCV13. The greatest benefit, again, was seen in AOM cost reductions. In the USA, using PCV20 contributed a USD 19.2 billion decline in direct costs, which largely came from averted AOM cases (5.4 million of the total 5.91 million PD cases) in a period of 10 years compared to PCV13 [49]. Also, the benefit was observed in reducing Invasive Pneumococcal (IPD) cases. In the United States, 21% of IPD cases were linked to PCV13, while PCV15 and PCV20 unique serotypes contributed to 17% and 22%, respectively, in children under five years of age [59]. It is worth mentioning that the percentage of IPD cases associated with serotypes covered by PCV15 and PCV20 was significantly higher than that of PCV13, by 1.8 and 2.9 times, respectively [59].

Furthermore, with 100% of studies showing dominant ICERs, it is pertinent to highlight that switching from PCV13 to PCV20 would be more beneficial in terms of direct cost reduction and yield a bigger gain in QALY compared to switching from PCV13 to PCV15. From a societal standpoint, the utilization of PCV20 is associated with noteworthy savings in indirect costs attributed to productivity loss in contrast to the employment of PCV15. Notably, in Canada, the adoption of PCV20 has facilitated savings exceeding CAD 300 million in indirect costs from a societal perspective, an amount equating to one-third of the averted direct costs, which amounted to CAD 1.5 billion (inflated to 2022 costs) [45]. Similarly, in the USA, PCV20 use saved USD 8 billion in direct costs, with work loss reductions of USD 1.9 billion, nearly a quarter of the total (adjusted to 2022 costs) [49].

PCV20′s superior impact on cost reduction compared to PCV15 was mostly caused by the more comprehensive serotype protection, specifically 10A, 11A, and 15B, which are expected to raise the incidence rate of IPD [53]. The economic burden of serotypes protected by licensed vaccines was investigated in a study conducted in thirteen countries that had established National Immunization Programs (NIPs) [60]. These countries included the United Kingdom, Australia, Austria, Canada, Finland, France, Germany, Italy, Netherlands, New Zealand, South Korea, Spain, and Sweden [60]. According to this investigation, PCV20 serotypes were responsible for an estimated 46% to 77% of pneumococcal infections. The nations presently adopting PCV13 in their NIPs (Australia, Canada, France, Germany, Italy, South Korea, Spain, and the United Kingdom) exhibited a disease burden ranging from 2% to 33% for PCV15-unique serotypes, whereas PCV20-unique serotypes contributed from 16% to 69% [60]. These findings highlight the more significant burden on PCV20 serotypes, suggesting that transitioning to PCV20 could result in more substantial cost reductions than PCV15.

The indirect effect played a crucial role in determining the cost-effectiveness of the pneumococcal vaccine. The indirect impact or herd effect refers to the protection provided to the unvaccinated population when a specific community achieves a high enough rate of vaccination uptake. Incorporating herd immunity into the economic analysis led to a reduction in disease transmission, contributing to a decrease in the number of cases of pneumococcal illness and, hence, lower treatment costs such as healthcare visits and hospitalization charges. Consequently, the ICER decreased, showing that the vaccination program becomes more cost-effective (less expensive, more efficient). This may explain why, in the scenario without herd effect, the ICERs of utilizing PCV13 were nearly 20 times higher in China than in the scenario with herd effect (79,304 CNY/QALY vs. 3777 CNY/QALY; the cost was inflated to 2015) as compared to no vaccination [56]. An analogous trend was noted in nations when comparing the cost-effectiveness of PCV15 vs. PCV13 [47,48,57]. PCV15 was shown to be a dominant strategy compared to PCV13 in Japan and the USA [47,48], as PCV15 contributed to more cost reduction and generated higher QALY gains. However, in the analysis conducted by Wilson et al. in the UK [57], the failure to consider the indirect effect resulted in an increase in vaccine and treatment costs, which caused a significant rise in the overall cost.

The PSA findings on PCV13 revealed that all simulations were concentrated in the northeast quadrant. This suggests that implementing PCV13 is more money-consuming and more effective than not vaccinating. In contrast, both PCV15 and PCV20 were located in the southeast quadrant, indicating that they were more cost-saving and showed greater effectiveness than PCV13. Therefore, both PCV15 and PCV20 are dominant compared to PCV13.

This is the first systematic review incorporating a cost-effectiveness analysis of PCV20 and PCV15. Given that the Advisory Committee on Immunization Practices (ACIP) suggested using two vaccinations with higher valency, PCV15 in 2022 and PCV20 in 2023, our findings can offer a comprehensive assessment of the costs associated with these vaccines. This information is crucial for determining which PCV should be included in the National Immunization Program (NIP). The majority of our selected evaluation covered the herd effect in 12 out of 16 cases, ensuring that the complete benefits of introducing PCVs may be adequately documented. Our study has multiple limitations. Each country exhibited varying levels of vaccine coverage, serotype distribution, and incidence rates of pneumococcal illness. Consequently, the ICER outcome may not accurately reflect the overall situation. Furthermore, the absence of clinical data about PCV15 and PCV20 necessitated the reliance on assumptions when considering the indirect impact on other serotypes covered by these vaccines.

In addition, differences in the health and healthcare systems of countries also significantly affect the factors that determine whether to use PCVs or not. For countries that have universal health coverage (UHC), it will be easier to support people than countries that do not. However, developed countries have much higher vaccine production costs than other countries. This is also a massive barrier to including vaccines in national immunization programs. Another factor is that copyright sharing in vaccine production is also a big barrier for most countries when pharmaceutical companies require a level of exclusivity or profits that are too high compared to the income of those countries.

The study also faced some limitations, as no meta-analysis of the studies reviewed in this study was conducted. In addition, the studies found in the search period from 2018 to 2024 were mainly conducted in Asia, with few studies in other regions of the world, which also affected the overall view of the country when adding a PCV to the general national immunization programs.

## 5. Conclusions

Both PCV13 and PCV10 were more cost-effective than no vaccination. In terms of higher-valency vaccines, PCV15 and PCV20 exhibited a notable decrease in direct costs and a greater increase in QALY compared to PCV13. Also, PCV20 was reported to be a dominant strategy compared to PCV15, regardless of vaccine schedule. With the advantages in terms of clinical factors as well as costs as indicated above, it can be seen that PCV is a vaccine worth considering for countries to include in their expanded immunization programs for vulnerable groups such as children—this is also the group that will help develop the country in the future.

## 6. Future Directions

As demonstrated above, PCVs have transformed the childhood immunization landscape, providing robust protection against a wide range of serious pneumococcal infections and reducing costs caused by the disease. In the future, we can expect continued advances in PCV technology, leading to even greater efficacy and durability. As vaccination coverage expands, the global burden of pneumococcal disease is expected to decline further. Furthermore, further research into personalized vaccines or combination vaccines promises to tailor immunization strategies to the needs of individual children.

## Figures and Tables

**Figure 1 healthcare-12-01950-f001:**
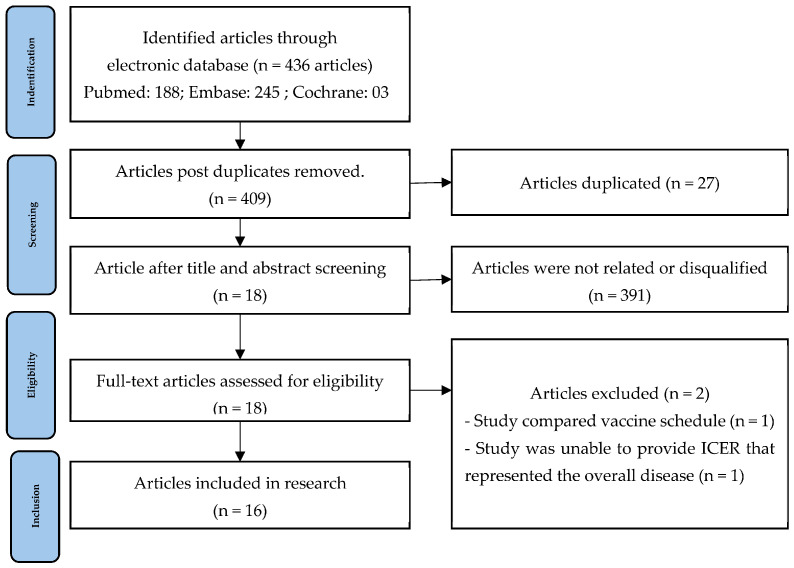
Flow diagram of studies selection.

**Table 1 healthcare-12-01950-t001:** Characteristics of selected articles.

No.	Author, Year, Country	Analysis Type	Intervention	Clinical Outcome	Model		Time Horizon	Discount Rate	Currency
1	Lytle et al., 2023, Canada [45]	CEA	PCV20 vs. PCV13	IPD	Markov		10 years	1.50%	2022 CAD
			PCV20 vs. PCV15	Pneumonia					
				AOM					
2	Sevilla et al., 2022, Egypt [42]	CBA	PCV13 vs. no vaccination	IPD	Markov		100 years	3%	2016 USD
		CUA	PCV10 vs. no vaccination	Pneumonia					
			PCV13 vs. PCV13	AOM					
3	Dilokthornsakul et al., 2019, Thailand [46]	CEA	PCV10 vs. no vaccination	IPD	Markov		Lifetime	3%	2018 TBH
			PCV10 vs. no vaccination	ACP					
				All cause-AOM					
4	Krishnamoorthy et al., 2019, India [51]	CEA	PCV13 vs. no vaccination	IPD	UNIVAC decision support		10 years	3%	2017 USD
				Pneumonia					
				AOM					
5	Shen et al., 2018, China [54]	CEA	PCV13 vs. no vaccination	IPD	Decision analytic model		1 year	3%	2015 CNY
				Pneumonia					
				AOM					
6	Dorji et al., 2018, Bhutan [43]	CUA	PCV13 vs. no vaccination	IPD	Markov		100 years	3%	2017 USD
			PCV10 vs. no vaccination	Pneumonia					
			PCV13 vs. PCV10	AOM					
7	Huang et al., 2023, USA [47]	CEA	PCV15 vs. PCV13	IPD	Markov		Lifetime	3%	2021 USD
				Pneumonia					
				AOM					
8	Tajima et al., 2023, Japan [48]	CEA	PCV15 vs. PCV13	IPD	Markov		10 years	2%	2015 USD
				NBPP					
				Pneumococcal AOM					
9	Li et al., 2021, China [56]	CEA	PCV13 vs. no vaccination	IPD	Decision analytic		1 year	5%	2019 CNY
				Pneumonia					
				AOM					
10	Wilson et al., 2022, UK [57]	CEA	PCV15 vs. PCV13	IPD	Economic model		5 years	3.50%	2021 GBP
			PCV20 vs. PCV13	Pneumonia					
			PCV20 vs. PCV15	AOM					
11	Chen et al., 2019, 180 countries * [55]	CEA	PCV13 vs. no vaccination	IPD	Decision tree		30 years	3%	2015 USD
				Pneumonia					
				AOM					
12	Perdrizet et al., 2021, Philippines [52]	CEA	PCV13 vs. PCV10-GSK	IPD	Decision analytic model		10 years	7%	2020 PHP
				Pneumonia					
				AOM					
13	Warren et al., 2023, Greece [53]	CEA	PCV20 vs. PCV15	IPD	Decision-analytic mode		10 years	3.50%	2023 EUR
				Pneumonia					
				AOM					
14	Huang et al., 2023, South Africa [44]	CUA	PCV13 vs. PCV10-GSK	IPD	Decision-analytic forecasting models		10 years	5%	2022 ZAR
			PCV13 vs. PCV10-SII	Pneumonia					
				AOM					
15	Rozenbaum et al., 2024, USA [49]	CEA	PCV20 vs. PCV13	IPD	Markov		10 years	3%	2022 USD
			PCV20 vs. PCV15	ACP					
				OM					
16	Ta et al., 2024, Germany [50]	CEA	PCV20 vs. PCV13	IPD	Markov		10 years	3%	2020 EUR
			PCV20 vs. PCV15	ACP					
				All-cause AOM					
**No.**	**Author, Year, Country**	**Analysis Type**	**Intervention**	**Clinical Outcome**	**Perspective**	**Vaccine Coverage**	**Funding**	**Health Outcome**	**SA**
1	Lytle et al., 2023, Canada [45]	CEA	PCV20 vs. PCV13 PCV20 vs. PCV15	IPD Pneumonia AOM	Payer Society	84%	Pfizer	QALY	DSA, PSA
2	Sevilla et al., 2022, Egypt [42]	CBA CUA	PCV13 vs. no vaccination PCV10 vs. no vaccination PCV13 vs. PCV13	IPD Pneumonia AOM	Society Payer	100%	Pfizer	RoR QALY, ICER	DSA, PSA
3	Dilokthornsakul et al., 2019, Thailand [46]	CEA	PCV10 vs. no vaccination PCV10 vs. no vaccination	IPD ACP All cause-AOM	Society	-	Pfizer	QALY, ICER	PSA
4	Krishnamoorthy et al., 2019, India [51]	CEA	PCV13 vs. no vaccination	IPD Pneumonia AOM	Government	88%	-	DALY, ICER	PSA
5	Shen et al., 2018, China [54]	CEA	PCV13 vs. no vaccination	IPD Pneumonia AOM	Payer	85%	Pfizer	QALY	DSA
6	Dorji et al., 2018, Bhutan [43]	CUA	PCV13 vs. no vaccination PCV10 vs. no vaccination PCV13 vs. PCV10	IPD Pneumonia AOM	Government	97%	WHO	QALY, ICER	DSA, PSA
7	Huang et al., 2023, USA [47]	CEA	PCV15 vs. PCV13	IPD Pneumonia AOM	Society	91.9%	Merck	QALY, LY, ICER	PSA, DSA
8	Tajima et al., 2023, Japan [48]	CEA	PCV15 vs. PCV13	IPD NBPP, Pneumococcal AOM	Payer Society	100%	Merck	QALY, ICER	PSA, DSA
9	Li et al., 2021, China [56]	CEA	PCV13 vs. no vaccination	IPD Pneumonia AOM	Payer	70%	-	QALY, ICER	DSA
10	Wilson et al., 2022, UK [57]	CEA	PCV15 vs. PCV13 PCV20 vs. PCV13 PCV20 vs. PCV15	IPD Pneumonia AOM	Payer	91%	Pfizer	QALY, LY, ICER	DSA
11	Chen et al., 2019, 180 countries * [55]	CEA	PCV13 vs. no vaccination	IPD Pneumonia AOM	Healthcare	-	WHO, Gavi, Bill & Melinda Gates Foundation	DALY, ICER	DSA, PSA
12	Perdrizet et al., 2021, Philipine [52]	CEA	PCV13 vs. PCV10-GSK	IPD Pneumonia AOM	Society	90%	Pfizer	LY, QALY, ICER	-
13	Warren et al., 2023, Greece [53]	CEA	PCV20 vs. PCV15	IPD Pneumonia AOM	Payer	84.5%	Pfizer	LY, QALY, ICER	PSA
14	Huang et al., 2023, South Africa [44]	CUA	PCV13 vs. PCV10-GSK PCV13 vs. PCV10-SII	IPD Pneumonia AOM	Payer	90.7%	Pfizer	LY, QALY, ICER	-
15	Rozenbaum et al., 2024, USA [49]	CEA	PCV20 vs. PCV13 PCV20 vs. PCV15	IPD ACP OM	Healthcare Society	83.5%	Pfizer	QALY, LYs	DSA, PSA
16	Ta et al., 2024, Germany [50]	CEA	PCV20 vs. PCV13 PCV20 vs. PCV15	IPD ACP All-cause AOM	Society	76.8%	Pfizer	LY, QALY, ICER	PSA, DSA

Abbreviation: CEA: Cost-Effectiveness Analysis; CUA: Cost-utility analysis; CBA: Cost-Benefit Analysis; IPD: Invasive Pneumococcal Disease; AOM: Acute Otitis Media; NBPP: Non-Bacteremic Pneumococcal Pneumonia; ACP: All-Cause Pneumonia; OM: Otitis Media; WHO: World Health Organization, Gavi: The vaccine alliance, QALY: Quality-Adjusted Life Years; DALY: Disability-adjusted life years; LY: Life Years; ICERs: Incremental Cost-Effectiveness Ratios; RoR: Rate of Return; SA: sensitivity analysis DSA: Deterministic sensitivity analyses; PSA: Probabilistic sensitivity analysis. *: Africa, Asia, Europe, Latin America and the Caribbean, North America, and Oceania.

**Table 2 healthcare-12-01950-t002:** Quality assessment following CHEERS 2022 statement.

No.	Item	Lytle et al., 2023 [45]	Sevilla et al., 2022 [42]	Dilokthornsaku et al., 2019 [46]	Krishnamoorthy et al., 2019 [51]	Shen et al., 2018 [54]	Dorji et al., 2018 [43]	Huang et al., 2023 [47]	Tajima et al., 2023 [48]
1	Title	1	1	1	1	1	1	1	1
2	Abstract	1	1	1	1	1	1	1	1
3	Background and objective	1	1	1	1	1	1	1	1
4	Health economic analysis plan	1	1	1	1	1	1	1	1
5	Study population	1	1	1	1	1	1	1	1
6	Setting and location	1	1	1	1	1	1	1	1
7	Comparators	1	1	1	1	1	1	1	1
8	Perspective	1	1	1	0.5	0.5	0.5	0.5	0.5
9	Time horizon	1	1	1	0.5	1	0.5	0.5	1
10	Discount rate	1	1	0.5	0.5	0.5	0.5	0.5	0.5
11	Selection of outcomes	1	1	1	1	1	1	1	1
12	Measurement of outcomes	1	1	1	1	1	1	1	1
13	Valuation of outcomes	1	1	1	1	1	1	1	1
14	Measurement and valuation of resources and costs	1	1	1	1	1	1	1	1
15	Currency, price date, and conversion	1	1	1	1	1	1	1	1
16	Rationale and description of model	1	1	1	1	1	1	1	1
17	Analytics and assumptions	0.5	1	1	1	1	1	0.5	0.5
18	Characterizing heterogeneity	0.5	1	1	1	1	0.5	1	1
19	Characterizing distributional effect	0.5	1	1	1	1	1	1	1
20	Characterizing uncertainty	1	1	1	1	1	1	1	1
21	Approach to engagement with patients and others affected by the study	0	0	0	0	0	0	0	0
22	Study parameters	1	1	1	1	1	1	1	1
23	Summary of main results	1	1	1	1	1	1	1	1
24	Effect of uncertainty	1	1	1	1	0.5	1	1	1
25	Effect of engagement with patients and others affected by the study	0	0	0	0	0	0	0	0
26	Study findings, limitations, generalizability, and current knowledge	1	1	1	1	1	1	1	1
27	Source of funding	1	1	1	1	1	1	1	1
28	Conflicts of interest	1	1	1	1	1	1	1	1
Total score		24.5	26	25.5	24.5	24.5	24	24	24.5
Conclusion		Good	Good	Good	Good	Good	Good	Good	Good
**No.**	**Item**	**Li et al., 2021** [56]	**Wilson et al., 2022** [57]	**Chen et al., 2019** [55]	**Perdrizet et al., 2021** [52]	**Warren et al., 2023** [53]	**Huang et al., 2023** [44]	**Rozenbaum et al., 2024** [49]	**Ta et al., 2024** [50]
1	Title	1	1	1	1	1	0.5	1	1
2	Abstract	1	1	1	1	1	1	1	1
3	Background and objective	1	1	1	1	1	1	1	1
4	Health economic analysis plan	1	1	1	1	1	1	1	1
5	Study population	1	1	1	1	1	1	1	1
6	Setting and location	1	0.5	1	0.5	1	1	1	1
7	Comparators	1	1	1	1	1	1	1	1
8	Perspective	0.5	0.5	0.5	0.5	0.5	1	1	1
9	Time horizon	1	0.5	0.5	0.5	0.5	0.5	1	1
10	Discount rate	0.5	0.5	0.5	1	0.5	0.5	1	1
11	Selection of outcomes	1	1	1	1	1	1	1	1
12	Measurement of outcomes	1	1	1	1	1	1	1	1
13	Valuation of outcomes	1	1	1	1	1	1	1	1
14	Measurement and valuation of resources and costs	1	1	1	1	1	1	1	1
15	Currency, price date, and conversion	0.5	1	1	1	1	1	1	1
16	Rationale and description of model	1	1	1	1	1	1	1	1
17	Analytics and assumptions	1	1	1	1	0.5	0.5	0.5	0.5
18	Characterizing heterogeneity	1	1	0.5	0.5	0	1	1	1
19	Characterizing distributional effect	1	0.5	0.5	0.5	0.5	0.5	0.5	0.5
20	Characterizing uncertainty	1	1	1	1	1	0	1	1
21	Approach to engagement with patients and others affected by the study	0	0	0	0	0	0	0	0
22	Study parameters	1	1	1	1	1	1	1	1
23	Summary of main results	1	1	1	1	1	1	1	1
24	Effect of uncertainty	1	1	1	1	1	0	0	1
25	Effect of engagement with patients and others affected by the study	0	0	0	0	0	0	0	0
26	Study findings, limitations, generalizability, and current knowledge	1	1	1	1	1	1	1	1
27	Source of funding	1	1	1	1	1	1	1	1
28	Conflicts of interest	1	1	1	1	1	1	1	1
Total score		24.5	23.5	23.5	23.5	22.5	21.5	24	25
Conclusion		Good	Good	Good	Good	Good	Good	Good	Good

**Table 3 healthcare-12-01950-t003:** Incremental effect in lower-valent pneumococcal vaccine.

Ref., Country, Currency	Schedule	Herd Effect	Vaccination Cost	Direct Cost	Indirect Cost	Total Cost	
PCV13 vs. No vaccination							
Sevilla et al., Egypt, 2016 USD [42]	2 + 1	Yes	43.63	−0.88	-	42.75	
Dorji et al., Bhutan, 2017 USD [43]	2 + 1	Yes	-	-	-	0.03	
Krishnamoorthy et al., India, 2017 USD [51]	2 + 1	No	35,000,000	−16,600,000	-	−16,600,000	
Dilokthornsakul et al., Thailand, 2018 TBH [46]	2 + 1 3 + 1	No	- -	- -	- -	2571 3693	
Shen et al., China, 2015 CNY [54]	3 + 1	No Yes	38,382,200,000 38,382,200,000	29,362,300,000 13,524,700,000	- -	29,362,300,000 13,524,700,000	
Li et al., China, 2019 CNY [56]	3 + 1	Yes	−323,757,862	−28,646,835	-	−28,646,835	
Chen et al., global, 2015 USD [55]	2 + 1, 3 + 1, 3 + 0	Yes	15,500,000,000	8,420,000,000	−2,640,000,000	6,670,000,000	
PCV10 vs. No vaccination							
Dilokthornsakul et al., Thailand, 2018 TBH [46]	2 + 1 3 + 1	No	- -	- -	- -	3881 5348	
Sevilla et al., Egypt, 2016 USD [42]	2 + 1	Yes	38.43	38.05	-	38.05	
Dorji et al., Bhutan, 2017 USD [43]	2 + 1	Yes	-	-	-	0.02	
PCV13 vs. PCV10							
Sevilla et al., Egypt, 2016 USD [42]	2 + 1	Yes	5.198	4.7	-	4.7	
Perdrizet et al., Philippines, 2020 PHP [52]	3 + 1	No	3,159,192,812	−1,399,247,136	−10,875,530,146	−12,274,777,282	
Huang et al., South Africa, 2022 ZAR [44]	2 + 1	No	587,690,427	−78,825,963	-	−78,825,963	
**Ref., Country, Currency**	**Schedule**	**Herd Effect**	**LYs**	**Effectiveness**	**ICER**	**CE Threshold**	**Cost-Effective**
PCV13 vs. No vaccination							
Sevilla et al., Egypt, 2016 USD [42]	2 + 1	Yes	-	0.0462 QALY	926	GDP: 3479	Yes
Dorji et al., Bhutan, 2017 USD [43]	2 + 1	Yes	-	0.0007 QALY	40	GDP: 2708	Yes
Krishnamoorthy et al., India, 2017 USD [51]	2 + 1	No	-	920,000 DALY	467	GDP: 1939.6	Yes
Dilokthornsakul et al., Thailand, 2018 TBH [46]	2 + 1 3 + 1	No	0.03 0.03	0.0349 QALY 0.0380 QALY	73,674 97,269	WTP: 160,000	Yes
Shen et al., China, 2015 CNY [54]	3 + 1	No Yes	- -	370,300 QALY 3,580,900 QALY	79,304 3777	GDP: 53,976	Yes Yes
Li et al., China, 2019 CNY [56]	3 + 1	Yes	-	14,880 QALY	Dominant	GDP: 157,300	Yes
Chen et al., global, 2015 USD [55]	2 + 1, 3 + 1, 3 + 0	Yes	-	9,130,000 DALY	724	WTP: 1000	Yes
PCV10 vs. No vaccination							
Dilokthornsakul et al., Thailand, 2018 TBH [46]	2 + 1 3 + 1	No	0.02 0.02	0.0228 QALY 0.0248 QALY	170,437 215,948	WTP: 160,000	No No
Sevilla et al., Egypt, 2016 USD [42]	2 + 1	Yes	-	0.0192 QALY	1984.414	GDP: 3479	Yes
Dorji et al., Bhutan, 2017 USD [43]	2 + 1	Yes	-	0.0006 QALY	36	GDP: 2708	Yes
PCV13 vs. PCV10							
Sevilla et al., Egypt, 2016 USD [42]	2 + 1	Yes	-	0.027 QALY	173.98	GDP: 3479	Yes
Perdrizet et al., Philippines, 2020 PHP [52]	3 + 1	No	156,061	153,349 QALY	Cost-saving	-	Yes
Huang et al., South Africa, 2022 ZAR [44]	2 + 1	No	4484	3191 QALY	Cost-saving	-	Yes

Abbreviation: WTP: Willingness-to-pay threshold; GDP: Gross Domestic Product; CE threshold: Cost-effectiveness threshold; LY: Life years; ICERs: Incremental Cost-Effectiveness Ratios; QALY: Quality-Adjusted Life Years; DALY: Disability-adjusted life years.

**Table 4 healthcare-12-01950-t004:** Incremental effect in higher-valent pneumococcal vaccine.

Ref., Country, Currency	Schedule	Herd Effect	Vaccine Cost	Direct Cost	Indirect Cost	Total Cost	
PCV15 vs. PCV13							
Huang et al., USA, 2021 USD [47]	3 + 1	Yes	25,200	−6,800,033,529	−4,017,519,577	−10,817,553,106	
Tajima et al., Japan, 2022 JPY [48]	3 + 1	Yes	3091	−235,135,797	−130,475,159	−365,610,955	
Wilson et al., UK, 2021 GBP [57]	1 + 1 vs. 1 + 1 2 + 1 vs. 1 + 1	No	7,900,205 212,402,154	1,124,922 200,554,981	-	1,124,922 200,554,981	
PCV20 vs. PCV13							
Lytle et al., Canada, 2022 CAD [45]	2 + 1	Yes	82,002,815	−3,226,480,346	−656,062,710	−3,882,543,056	
Wilson et al., UK, 2021 GBP [57]	1 + 1 vs. 1 + 1 2 + 1 vs. 1 + 1	No	38,303,366 215,602,573	−459,192,688 −403,126,911	-	−459,192,688 −403,126,911	
Rozenbaum et al., USA, 2022 USD [49]	3 + 1	Yes	2,338,463,867	−19,189,701,809	−3,726,859,511	−20,578,097,453	
Ta et al., Germany, 2022 EUR [50]	3 + 1 vs. 2 + 1	Yes	525,362,283	−2,035,127,528	−358,136,083	−2,393,263,611	
PCV20 vs. PCV15							
Lytle et al., Canada, 2022 CAD [45]	2 + 1	Yes	82,083,788	−1,484,267,884	−307,853,576	−1,792,121,460	
Wilson et al., UK, 2021 GBP [57]	1 + 1 vs. 1 + 1 1 + 1 vs. 2 + 1 2 + 1 vs. 2 + 1 2 + 1 vs. 1 + 1	No	30,403,161 −174,098,788 3,200,419 207,702,386	−460,317,610 −659,747,669 −603,681,892 −404,251,833	-	−460,317,610 −659,747,669 −603,681,892 −404,251,833	
Warren et al., Greece, 2023 EUR [53]	3 + 1	No	−4,566,825	−58,138,419	-	-58,138,419	
Rozenbaum et al., USA, 2022 USD [49]	3 + 1	Yes	2,437,771,654	−8,003,928,578	−1,898,767,496	− 9,902,696,074	
Ta et al., Germany, 2022 EUR [50]	3 + 1 vs. 2 + 1	Yes	522,747,819	−1,343,839,409	−284,161,097	-1,628,000,506	
**Ref., Country, Currency**	**Schedule**	**Herd Effect**	**LYs**	**Effectiveness**	**ICER**	**CE Threshold**	**Cost-Effective**
PCV15 vs. PCV13							
Huang et al., USA, 2021 USD [47]	3 + 1	Yes	90,026	96,056 QALY	Dominant	-	Yes
Tajima et al., Japan, 2022 JPY [48]	3 + 1	Yes	7	24 QALY	Dominant	-	Yes
Wilson et al., UK, 2021 GBP [57]	1 + 1 vs. 1 + 1 2 + 1 vs. 1 + 1	No	262 475	361 QALY 640 QALY	3112 313,229	WTP: 20,000	Yes No
PCV20 vs. PCV13							
Lytle et al., Canada, 2022 CAD [45]	2 + 1	Yes	-	47,056 QALY	Dominant	-	Yes
Wilson et al., UK, 2021 GBP [57]	1 + 1 vs. 1 + 1 2 + 1 vs. 1 + 1	No	23,165 28,818	28,096 QALY 35,009 QALY	Dominant Dominant	WTP: 20,000	Yes Yes
Rozenbaum et al., USA, 2022 USD [49]	3 + 1	Yes	515,203	271,414 QALY	Dominant	-	Yes
Ta et al., Germany, 2022 EUR [50]	3 + 1 vs. 2 + 1	Yes	563,014	904,854 QALY	Dominant	-	Yes
PCV20 vs. PCV15							
Lytle et al., Canada, 2022 CAD [45]	2 + 1	Yes	-	21,881 QALY	Dominant	-	Yes
Wilson et al., UK, 2021 GBP [57]	1 + 1 vs. 1 + 1 1 + 1 vs. 2 + 1 2 + 1 vs. 2 + 1 2 + 1 vs. 1 + 1	No	22,903 22,690 28,343 28,556	27,735 QALY 27,456 QALY 34,369 QALY 34,648 QALY	Dominant Dominant Dominant Dominant	WTP: 20,000	Yes Yes Yes Yes
Warren et al., Greece, 2023 EUR [53]	3 + 1	No	551	486 QALY	110,000	-	Yes
Rozenbaum et al., USA, 2022 USD [49]	3 + 1	Yes	279,655	146,168 QALY	Dominant	-	Yes
Ta et al., Germany, 2022 EUR [50]	3 + 1 vs. 2 + 1	Yes	400,731	646,235 QALY	Dominant	-	Yes

Abbreviation: WTP: willingness-to-pay threshold; GDP: Gross Domestic Product; CE threshold: cost-effectiveness threshold; LY: life years; ICERs: Incremental Cost-Effectiveness Ratios; QALY: Quality-Adjusted Life Years; DALY: Disability-adjusted Life Years.

**Table 5 healthcare-12-01950-t005:** Sensitivity analysis in lower-valent pneumococcal vaccines.

Ref.	DSA	PSA	
	The Most Impactful Parameter on ICERs	Probability	Quadrant
PCV13 vs. no vaccination			
Sevilla et al. [42]	-Base-year incidence rates -Discount rate -PCV direct and indirect effects on in-patient pneumonia -Modeling horizon length	-	-
Dilokthornsakul et al. [46]	-	100%	Northeast
Krishnamoorthy et al. [51]	-	100%	Northeast
Shen et al. [54]	Incidence rates of inpatient pneumonia in ages 0–4	-	-
Dorji et al. [43]	-The variation in serotype coverage -Duration of vaccine protection -Excluding indirect vaccine effects (herd protection) -Discount rate	-	-
Li et al. [56]	-Incidence of inpatient pneumonia 0–2 y, 2–4 y, 18–34 y -Total direct cost -Discount rate	-	-
Chen et al. [55]	-Disease incidence -Case fatality rate -Vaccine price	100%	Northeast
PCV10 vs. no vaccination			
Sevilla et al. [42]	-Base-year incidence rates -Discount rate -PCV direct and indirect effects on inpatient pneumonia -Modeling horizon length	-	-

Abbreviation: ICERs: Incremental Cost-Effectiveness Ratios; DSA: Deterministic sensitivity analysis; PSA: Probabilistic sensitivity analysis; PCV: pneumococcal conjugate vaccine.

**Table 6 healthcare-12-01950-t006:** Sensitivity analysis in higher-valent pneumococcal vaccines.

Ref.	DSA		PSA	
	Interest Value	Most Impactful Parameter	Probability	Quadrant
PCV20 vs. PCV13				
Lytle et al. [45]	Cost	-Percentage of the indirect effect of PCV20 accrued -The steady-state indirect effects against hospitalized pneumonia -Age-specific serotype distribution of hospitalized pneumonia -The direct medical cost per hospitalized pneumonia episode	100%	Southeast
	QALY	-Utility decrement of simple OM -Utility decrement of hospitalized pneumonia -Utility decrement of non-hospitalized pneumonia		
Wilson et al. [57]	NMB	-Percentage PP cases (≥65 years) -The hospitalized pneumonia incidence (≥65 years) -The direct costs for hospitalized pneumonia (≥65 years)	-	-
Rozenbaum et al. [49]	Cost	-Vaccine serotype coverage -Indirect effect accrual for PCV20 -PCV20 and PCV13 cost per dose	100%	Southeast
	QALY	-Indirect effect accrual for PCV20 -Vaccine serotype coverage -Maximum indirect effect for all-cause hospitalized NBP		
Ta et al. [50]	Cost	-Maximum indirect effect against hospitalized pneumonia (PCV20) -Serotype distribution by age -Incidence of hospitalized pneumonia -Cost per episode of hospitalized pneumonia	100%	Southeast
	QALY	-Maximum indirect effects on hospitalized pneumonia (PCV20) -Serotype distribution by age -Baseline utilities -Hospitalized pneumonia incidence -CFR for hospitalized pneumonia		
PCV15 vs. PCV13				
Huang et al. [47]	ICERs	-VEs against all-cause inpatient pneumonia -Vaccine coverage rate -Indirect effects -Incidence and fatality rates of bacteremic pneumonia in the elderly	100%	Southeast
Tajima et al. [48]	ICERs	-PCV15 and PCV13 serotype-specific VE in in-patient pneumonia (including serotype-specific VE for V114 and PCV13) -Direct and indirect cost per episode -Baseline incidence rate -Percentage attributable to *S. pneumoniae* -Serotype distribution -QALY decrement	98.7%	Southeast
PCV20 vs. PCV15				
Warren et al. [53]	-	-	100%	Southeast
Rozenbaum et al. [49]	Cost	-Indirect effect accrual for PCV20 -Cost per dose of PCV20 and PCV15 -Maximum indirect effect in hospitalized pneumonia for PCV20 -Vaccine serotype coverage	100%	Southeast
	QALY	-Indirect effect accrual for PCV20 -Maximum indirect effect in hospitalized pneumonia for PCV20 -Indirect effect accrual for PCV15 -Vaccine serotype coverage		
Ta et al. [50]	Cost	-Maximum indirect effect against hospitalized pneumonia (PCV20) -Serotype distribution by age -Incidence of hospitalized pneumonia -Cost per episode of hospitalized pneumonia	98.4%	Southeast
	QALY	-Maximum indirect effects on hospitalized pneumonia (PCV20) -Serotype distribution by age -Baseline utilities -Hospitalized pneumonia incidence -Indirect effect accrual for PCV20		

Abbreviation: ICERs: Incremental Cost-Effectiveness Ratios; DSA: Deterministic sensitivity analysis; PSA: Probabilistic sensitivity analysis; PCV: pneumococcal conjugate vaccine; QALY: Quality-Adjusted Life Years; OM: otitis media; PP: pneumococcal pneumonia; VE: vaccine effectiveness; CFR: case fatality rate, NMB: net monetary benefit, NBP: all-cause non-bacteremic pneumonia.

## Data Availability

Data were collected from PubMed, Embase, and Cochrane databases.

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
