# Peer review of "Cost-Effectiveness Analysis of Pneumococcal Vaccines in the Pediatric Population: A Systematic Review"

_healthcare, 2024, doi:10.3390/healthcare12191950_

Round 1

Reviewer 1 Report

Comments and Suggestions for Authors

Dear authors, Thank you for conducting this important study on vaccines and their impact on disease prevention. I have some questions and concerns regarding your research:

1) As you collected studies that were published during the COVID-19 pandemic, do you think external factors related to the pandemic may have influenced the findings of these studies? If so, how do you account for this potential impact in your analysis?

2) It seems that your study is focused on cost-effectiveness, but the data collected includes cost-utility and cost-benefit analyses as well. Can you clarify how these different approaches are being integrated into your study?

Thanks

Reviewer 2 Report

Comments and Suggestions for Authors

I have reviewed your manuscript "Cost-effectiveness analysis of pneumococcal vaccine in the pediatric population: A systematic review" with great interest. Congratulations on completing and submitting this comprehensive work. Your systematic review provides a valuable contribution to our understanding of the economic impact of pneumococcal conjugate vaccines.

The strengths of your study are evident. Your inclusion of newer vaccines is particularly timely and offers a comprehensive global perspective. The consideration of herd immunity effects is commendable.

However, there are several areas where the manuscript could be improved:

  1. The introduction, while informative, is quite lengthy. Consider streamlining it to enhance readability and focus.
  2. Your search strategy appears somewhat limited. Expanding your search terms and databases might yield additional relevant studies.
  3. Some of the data tables are complex and could benefit from simplification or alternative presentation methods to enhance clarity.
  4. The discussion of cross-country comparisons requires more depth. Healthcare systems, vaccine pricing, and distribution processes vary significantly between countries. A more nuanced analysis of how these factors might influence your conclusions would strengthen the paper considerably.
  5. The limitations section would benefit from expansion. A more thorough discussion of potential biases, methodological constraints, and challenges in data interpretation would demonstrate a deeper critical engagement with your findings.
  6. A brief section on the practical implications of your findings for healthcare providers and policymakers would enhance the paper's relevance and impact.

These suggestions notwithstanding, your work represents a significant contribution to the field. With some revisions, I believe it has the potential to be an influential paper in guiding vaccination policies.

Reviewer 3 Report

Comments and Suggestions for Authors

The manuscript is well structured and the results are presented in a very concise and logical way. I would recommend adding to the methodology section the initials of the researchers who did the initial screening, assessment against inclusion/exclusion criteria, etc. Otherwise, I have not any other recommendations.

Author Response

Dear Reviewer 03,

On behalf of my co-authors, I express my deep gratitude and appreciation for taking the time to contribute to our manuscript, “Cost-effectiveness analysis of pneumococcal vaccine in the pediatric population: A systematic review” which will be considered for publication in Healthcare.

For your comments, the author updated it in the search strategy part of the Materials and Methods part.

Once again, thank you very much. If you have any questions, please feel free to contact me.